# The Role of the Cyanobacterial Type IV Pilus Machinery in Finding and Maintaining a Favourable Environment

**DOI:** 10.3390/life10110252

**Published:** 2020-10-23

**Authors:** Fabian D. Conradi, Conrad W. Mullineaux, Annegret Wilde

**Affiliations:** 1School of Biological and Chemical Sciences, Queen Mary University of London, London E1 4NS, UK; f.d.conradi@qmul.ac.uk (F.D.C.); c.mullineaux@qmul.ac.uk (C.W.M.); 2Institute of Biology III, University of Freiburg, Schänzlestr. 1, 79104 Freiburg; Germany

**Keywords:** Type IV pili, cyanobacteria, phototaxis, flocculation, competence, *Synechocystis*

## Abstract

Type IV pili (T4P) are proteinaceous filaments found on the cell surface of many prokaryotic organisms and convey twitching motility through their extension/retraction cycles, moving cells across surfaces. In cyanobacteria, twitching motility is the sole mode of motility properly characterised to date and is the means by which cells perform phototaxis, the movement towards and away from directional light sources. The wavelength and intensity of the light source determine the direction of movement and, sometimes in concert with nutrient conditions, act as signals for some cyanobacteria to form mucoid multicellular assemblages. Formation of such aggregates or flocs represents an acclimation strategy to unfavourable environmental conditions and stresses, such as harmful light conditions or predation. T4P are also involved in natural transformation by exogenous DNA, secretion processes, and in cellular adaptation and survival strategies, further cementing the role of cell surface appendages. In this way, cyanobacteria are finely tuned by external stimuli to either escape unfavourable environmental conditions via phototaxis, exchange genetic material, and to modify their surroundings to fit their needs by forming multicellular assemblies.

## 1. The Type IV Pilus Machinery Conveys Twitching Motility to Cyanobacteria

Cyanobacteria are found in a wide variety of ecological niches, ranging from polar latitudes [1,2] to desert soil crusts [3]. In marine environments, a small number of cyanobacterial genera are so abundant that they account for a substantial portion of marine primary productivity [4]. Despite the differences in preferred niches and cellular morphology, many species of cyanobacteria take a remarkably similar approach in adapting to environmental changes. Cyanobacteria are not known to possess flagella and instead, most motile cyanobacteria rely on T4P to convey twitching motility across surfaces, allowing them to move towards favourable environments or to escape unfavourable environments. A well-studied example of cyanobacterial twitching motility is found in *Synechocystis* sp. PCC 6803 (hereafter *Synechocystis*), but other instances have been reported and characterised in both single-celled [5] and filamentous cyanobacteria [6], which largely match the *Synechocystis* T4P machinery in component genes and operon structure [7].

T4P are protein filaments extended from a membrane-spanning pore complex. In *Synechocystis*, the components of the T4P complex have been largely identified by homology with the Type IVa pilus (T4aP) systems of the heterotrophic Gram-negative bacteria *Myxococcus xanthus* [8] and *Pseudomonas aeruginosa* [9], in contrast to the Type IVb pilus, which mainly differs by the length and the amino acid sequence of the pilin proteins [10]. The filament, composed of pilin proteins characterised by their cleavage motif [11], is extended by the secretion ATPase PilB and passes through the membrane at the inner membrane platform protein PilC and the outer membrane pore PilQ. PilC and PilQ are connected by a set of accessory proteins. The structure and function of the cyanobacterial T4P apparatus have been reviewed in [7]. A schematic representation of the T4P complex is given in Figure 1.

Electron micrographs show two types of cell appendages on the *Synechocystis* surface, which have been termed thick pili and thin pili for their respective diameters. Thick *Synechocystis* pili have a diameter of 6–8 nm [8], matching the diameter of T4P found in heterotrophic bacteria [17]. Thick pili are essential for twitching motility and deletion mutants of the *pilA1* gene encoding the major pilin, a homologue of *P. aeruginosa* PilA, lose both thick pili and twitching motility. It is currently unclear whether PilA1 is the only pilin incorporated in *Synechocystis* thick pili. In contrast, thin pili are 2–3 nm in diameter and unable to convey twitching motility by themselves, as Δ*pilA1* mutants retain thin pili [8]. The nature and subunit composition of *Synechocystis* thin pili have not been solved so far. However, recent structures of *Thermus thermophilus* wide and narrow pili have shown that T4P machinery can produce structurally distinct pili depending on the type of pilin subunit incorporated [18]. Besides PilA1, *Synechocystis* contains a complement of ten other known PilA variants termed minor pilins [11]. The filamentous cyanobacterium *Nostoc punctiforme* also contains multiple PilA-like proteins and many of the components of the *N. punctiforme* T4P machinery have been identified according to *Synechocystis* annotations [19].

Pilins are processed by the PilD peptidase, which cleaves the N-terminal signal peptide. The mature pilins are then polymerised by the action of the hexameric ATPase PilB [20], thought to rotate during ATP hydrolysis, rotating PilC in turn and incorporating pilin subunits [21]. The retraction motor PilT, also a hexameric ATPase, rotates in the opposite direction to PilB, depolymerising the filament during pilus retraction [21]. The polymerised filament exits the outer membrane through PilQ. A set of accessory proteins have been identified in *Synechocystis* by homology with heterotrophic T4P systems, termed PilM, PilN, and PilO [9]. A PilP homologue has recently been identified in *Synechococcus elongatus* PCC 7942 [16]. Taton et al. also suggest that several other cyanobacteria, including *Synechocystis*, encode PilP homologues, though the sequence conservation is less pronounced [16]. As shown in Figure 1, the alignment complex composed of the PilMNOP accessory proteins forms a set of two rings in the periplasmic space [12] and links the T4P apparatus components in the inner and outer membranes. A recent study also indicated that many marine picocyanobacterial strains may contain a T4P apparatus homologous to *S. elongatus* PCC 7942 [22].

In contrast to single-celled motile cyanobacteria like *Synechocystis* or *Synechococcus elongatus* UTEX 3055, vegetative filaments of many filamentous cyanobacteria are not inherently motile. Instead, they differentiate into specialised motile filaments termed hormogonia. While vegetative filaments of *N. punctiforme* are non-piliated, cells that are part of hormogonia show abundant surface piliation [19] arrayed in rings at the cell poles [23] and are capable of twitching motility using T4P [6]. In contrast to *Synechocystis*, however, only thick pili have been detected in *Nostoc* hormogonia [19].

## 2. Twitching Motility Enables Cyanobacteria to Seek out Favourable and Escape Unfavourable Environments

Cyanobacteria couple twitching motility conveyed by the T4P apparatus with environmental sensing and engage in tactic behaviour—the movement towards or away from light sources (phototaxis) or chemical gradients (chemotaxis). Phototaxis allows cyanobacteria to move towards environments that provide sufficient photosynthetically active light, avoiding higher light intensities or shorter wavelengths of light that might damage the photosynthetic apparatus [24]. Chemotaxis has been reported for some species of filamentous cyanobacteria such as *N. punctiforme* in response to plant hosts [25], which provide opportunity for symbiotic lifestyles, and *Oscillatoria* towards CO_2_ [26]. Twitching motility-based phototaxis, in contrast, is a much more common feature in cyanobacteria, being observed and studied in single-celled cyanobacteria such as *Synechocystis* [27] and *S. elongatus* UTEX 3055 [5] as well as filamentous cyanobacteria [23]. The means by which phototaxis is performed, however, varies between cyanobacteria. Filamentous cyanobacteria like *N. punctiforme* perform both chemotaxis and phototaxis by moving up and down gradients, the motile hormogonia moving along the filament axis and controlling the rates of reversal in order to achieve net motion towards the attractant [6]. When applied to phototaxis, this behaviour is termed scotophobia [28]. Light perception in *Phormidium uncinatum*, another filamentous cyanobacterium, depends on decreases in light intensity at the leading end of the filament and increases in light intensity at the lagging end, though the mechanism for this spatial comparison is unclear [29]. Despite the prevalence of movement aligned with the filament axis, partial illumination of *Anabaena variabilis* trichomes has produced motility towards a light source placed to the side of the filament, indicating some ability to reorient the trichome and thus, the direction of movement [30]. Until recently, the mechanism by which motile filamentous cyanobacteria such as *N. punctiforme*, *Oscillatoria* [31], or *Anabaena variabilis* [32] move was under debate, with other hypotheses besides T4P including polysaccharide extrusion as the driving force of hormogonium motility [33]. However, it has become clear that at least *N. punctiforme* hormogonia move using a T4P system with the pili exerting a pulling force [6], as is common for twitching motility. The hormogonium polysaccharides extruded by *N. punctiforme* nonetheless support motility across surfaces [23] as has also been suggested for *Synechocystis* [34,35,36]. The hormogonium achieves the coordinated T4P action required for this behaviour by dynamically localising the partial coiled-coil-rich protein HmpF, encoded by a gene in the chemotaxis-like *hmp* gene cluster, to the forward cell pole [6]. Cho et al. showed that HmpF is essential for hormogonium motility and phototaxis as well as cell piliation in *N. punctiforme* [6].

In contrast to the bias towards a light source in *N. punctiforme*, *Synechocystis* has been shown to sense light directly by using the cell body as a microlens [37,38]. Although several control elements which are likely to be involved in downstream signal transmission have been identified, the exact signal transduction mechanism is unclear so far. The lensing of light in *Synechocystis* enables the cells to perform true phototaxis, moving directly towards light sources instead of performing biased random walks as in chemotaxis. *Synechocystis* has pili distributed around the entire cell perimeter and dynamically localises the extension motor PilB1 in the direction of movement [39], whereas *N. punctiforme* do not show dynamic PilB localisation [23]. This feature enables *Synechocystis* to move in any direction rather than being constrained in the direction to the cell or filament axis.

The regulation of twitching motility in response to external stimuli is achieved by sets of receptors and downstream regulators. In *Synechocystis*, while light lensing through the cell body provides the directionality to motility, movement towards or away from the light source is determined by the action of several photoreceptors. In particular, excess blue and UV light cause lower growth rates in *Synechocystis* [40], induce non-photochemical quenching in photosynthetic organisms indicating photoinhibition [41], oxidative stress [42], and various other deleterious effects in different cyanobacteria [43]. Many *Synechocystis* photoreceptors thus perceive short-wavelength light and *Synechocystis* cells use this information to either cease movement (blue light) [44] or reverse movement direction and move away from the stimulus (UV-A light) [45]. Stimuli from longer-wavelength light are also sensed, such as high-intensity red light leading to no movement or even negative phototaxis [46,47]. Cyanobacteriochromes (CBCR), Light-oxygen-voltage (LOV) photoreceptors, and Sensor of Blue Light using Flavin adenine dinucleotide (BLUF) photoreceptors sense the wavelength and intensity of incident light by photoconversion of bound bilin [48], flavin mononucleotide [49], or flavin adenine dinucleotide [50] chromophores, respectively.

The CBCR PixJ1 controls the direction of motility depending on the ratio of blue/green light [51] and mutants deficient in PixJ1 perform negative phototaxis (movement away from the light source) in environments where wt *Synechocystis* move towards the light source [52]. PixJ1 is part of the chemotaxis-like system *tax1* and carries a methyl-accepting chemotaxis protein (MCP)-like domain [51] and the downstream CheY-like response regulator PixG contains a PATAN domain thought to interact with the T4P apparatus [28,53]. PixJ homologues have been identified in various cyanobacteria, including *Nostoc punctiforme* ATCC 29133 [54], *Thermosynechococcus elongatus* BP-1 [55], *Anabaena* sp. PCC 7120 [56], and *Synechococcus elongatus* UTEX 3055 [5]. The photoconversion of the different PixJ homologues is variable across a wide range including red/green [57] and blue/green photocycles [51]. However, only some of the species confirmed to carry PixJ homologues have been shown to use them in phototaxis [5,54], partially owed to the general loss of motility in several cyanobacterial isolates. UirS, another *Synechocystis* CBCR, contains a UV/green photoconvertible GAF domain and causes motility reversal under UV illumination via its downstream effector, the PATAN domain containing response regulator LsiR [45]. Although the mechanism of action by which LsiR affects the T4P apparatus is not currently known, the dependence of negative phototaxis under UV-A illumination on both *uirS* and *lsiR* indicates their interaction with the T4P machinery either directly or indirectly. A third photoreceptor system which is involved in controlling phototaxis in *Synechocystis* is the PixD–PixE complex [58]. PixD is a BLUF domain photoreceptor which binds the PATAN domain response regulator PixE [59,60]. It is hypothesised that upon blue light exposure, PixE dissociates from PixD, binds to PilB1 and reverses the direction of movement, resulting in negative phototaxis [61]. The *Synechocystis* chemotaxis-like system *tax3* for which the signal input is unknown, leads to loss of thick pili and motility entirely when disrupted [52,62]. This system also contains a PATAN domain CheY-like response regulator, implying that PATAN domains could link external signals with the T4P apparatus to control its function or localization. The photoreceptor Cph2 is also implicated in *Synechocystis* motility. Mutants deficient in Cph2 show phototaxis towards blue light, whereas wt *Synechocystis* are non-motile in the same conditions [44]. In contrast to PixJ1, Cph2 transmits downstream signals through the production of the second messenger cyclic di-GMP (c-di-GMP) via its C-terminal GGDEF domain, which becomes active when blue light intensity is high relative to green light intensity [63]. Elevated c-di-GMP levels are commonly associated with reduced motility and increased sessility (reviewed in [64]). The blue/green light-dependent activity of Cph2 also leads to a host of pilus- and cell surface-related transcriptional changes [65], indicating multiple modes of action of Cph2-based c-di-GMP signalling. Furthermore, the N-terminus of Cph2 contains a red/far-red light sensing dual GAF domain module. The c-di-GMP synthesizing enzyme Slr1143 interacts with Cph2 and modulates motility under high-intensity red light by a so far unknown mechanism [46]. In *S. elongatus* PCC 7942, the LOV domain-based blue light receptor SL2 shows phosphodiesterase activity, breaking down c-di-GMP in the dark. This process unexpectedly accelerates upon blue illumination [66]. As *S. elongatus* PCC 7942 is non-motile, however, the effect of SL2-based photoperception on motility cannot be assessed, although the closely related and motile *Synechococcus elongatus* UTEX 3055 strain [5] may enable studies of the involvement of photoreceptors in *Synechococcus* phototaxis and biofilm formation. The mechanisms of many of the photoreceptors mentioned here have recently been reviewed in [28].

Cyanobacteria thus move towards light sources until one or more photoreceptors detect unfavourable light conditions, such as high-intensity red, blue, or UV light, which provoke a change in movement direction or a cessation of motility. In this way, cyanobacteria can position themselves in an optimal light environment. A model of this process in a biofilm context is shown in Figure 2. A biofilm provides a substrate for T4P to latch onto during twitching motility, making it a more likely environment for applying such a model compared to planktonic culture. The variety of chromophores, photosensing domains, and downstream signalling mechanisms with which cyanobacteria regulate the direction and extent of phototaxis via the T4P apparatus enable a high degree of adaptation to fluctuating environmental light conditions.

## 3. Many Species of Cyanobacteria Form Large-Scale Multicellular Assemblages

While phototaxis and chemotaxis enable cyanobacteria to seek out favourable environments, many species of cyanobacteria are also known to form multicellular assemblies, allowing cyanobacteria to create their own niches. Aggregate formation in bacteria usually involves the secretion of various substances such as polysaccharides, proteins, and nucleic acids, which together form extracellular polymeric substances (EPS). Examples of aggregate formation include the well-documented blooms of floating *Microcystis aeruginosa* colonies, a unicellular freshwater cyanobacterium showing extensive cell surface piliation in liquid medium and on agar surfaces [68]. Nakasugi and Neilan (2005) present electron micrographs which indicate connections between cells that are likely made by T4P [68]. Colonies of different *Microcystis* species vary in size and growth rate with nutrient concentrations and growth medium pH [69]. Ma et al. (2014) suggest the smaller colony sizes and higher growth rate observed in certain high N and high P conditions represent a better utilisation of the available nutrients by *Microcystis* [69]. Nutrient starvation in bacterial aggregates has also been observed in heterotrophs [70] and *Synechocystis* [71] and slow mass-transfer into aggregate interiors is one of the drawbacks of communal lifestyles, leading to steep gradients of important nutrients such as CO_2_ [72]. Similarly to *M. aeruginosa*, *Synechocystis* cell growth increased and lower aggregation was observed when the extracellular nutrient concentration was raised [71].

*Synechocystis* cells form floating aggregates termed flocs [71,73], although these flocs show a less dense colony morphology than their *Microcystis* counterparts. They are distinguished from biofilms in their lack of attachment to a substratum. Formation of these filamentous structures is dependent on some T4P components and the string-like appearance of *Synechocystis* flocs (shown in Figure 3b,c) might be a strategy to minimise nutrient limitation by increasing the surface-area-to-volume ratio [71]. Despite the disadvantages of nutrient limitation experienced in colonial lifestyles, lower diffusion rates through EPS also benefit the cells contained in it. Heterotrophic bacteria such as *P. aeruginosa* [74] and *Klebsiella pneumoniae* [75] show significantly increased resilience to many antibiotics in intact biofilms compared to planktonic cultures. Although the non-infectious cyanobacteria are less likely to encounter antibiotic treatment, the same principle of EPS shielding cells from harmful extraneous effects has been extended to salt and metal ion toxicity [76,77,78], phage infection [79], and predation by some (though not all) grazers [80]. Work on the *Pseudomonas* genus has shown that functional, retractile T4P are also required for infection by a number of bacteriophages [81,82,83]. Although this has not been confirmed to date in cyanobacteria to our knowledge, particularly as no known phages exist for the frequently studied *Synechocystis*, it may provide an additional layer of phage protection in multicellular aggregate contexts where a high intracellular c-di-GMP level down-regulates T4P dynamics.

The flocculation process in *Synechocystis* is known to be dependent on light colour via Cph2-based c-di-GMP signalling, increasing flocculation in blue light relative to green light [71]. *Synechocystis* biofilm formation has likewise been shown to be stimulated by blue light, likely via c-di-GMP [84]. Similar wavelength-dependent effects have been described for aggregation in *Thermosynechococcus vulcanus*, where the blue/green photoconvertible CBCR SesA induces aggregation in blue/UV light [85,86]. The SesB and SesC CBCRs in turn suppress *T. vulcanus* aggregation in red or green light [86]. Aggregate formation in *Synechocystis* and *Thermosynechococcus* may thus be, among other functions, a protective measure against short-wavelength light, both by increased light attenuation in dense parts of cyanobacterial aggregates [72] and by secretion of photoprotective extracellular matrix components as seen in various cyanobacteria [43,87,88]. It has been proposed that particularly the regulation of aggregation in response to the ratio of blue to green light by photoreceptors like Cph2 or the SesABC system might also serve to sense cell shading and regulate the size of aggregates, as green light penetrates deeper into cyanobacterial aggregates than blue light [71,89]. Enomoto and Ikeuchi [89] have shown that *T. vulcanus* aggregation is dependent on initial culture density as would be expected for such a system. A proposed model of this process is shown in Figure 3a. Some species of filamentous cyanobacteria are known to move vertically within phototrophic mats with diurnal cycles and in response to different light intensities [67,90] and wavelengths [91]. Cell shading sensors like blue/green photoconvertible photoreceptors of *Synechocystis* and *Thermosynechococcus* [89] and direct light direction sensing in *Synechocystis* may also enable cyanobacteria in complex communities to determine their location within the community [37]. However, to our knowledge, little evidence exists of unicellular cyanobacterial genera taking part in such migrations to date. Figure 2 shows a speculative model of such migrations according to light gradients for single cellular cyanobacteria in a multi-species phototrophic mat. In this model, cyanobacteria migrate in the direction of incident light towards the surface of the biofilm until blue light intensity relative to green light intensity is sufficient to block further motility. Cyanobacteria near the surface of the biofilm would in turn migrate away from UV-A radiation, which is strongly attenuated by the biofilm [91]. In this way, a zone might form in phototrophic mats in which cyanobacteria accumulate, with precise depth likely depending on the species. Cyanobacteria could thus control their light environment even within communities. This model comes with the caveat of a lack of spatiotemporal data on motility and c-di-GMP levels in aggregates of unicellular cyanobacteria.

Another example of unicellular cyanobacteria forming multicellular communities is *S. elongatus* PCC 7942 biofilm formation. Intriguingly, *S. elongatus* PCC 7942 possesses an autoinhibitory system controlling biofilm formation in a manner dependent on a PilB homologue and PilC [92]. Schatz et al. (2013) found that when the T4P system was disrupted at the point of the motor ATPase or PilC, the usually planktonic *S. elongatus* PCC 7942 formed biofilms at the bottom of culture vessels [92].

## 4. The T4P Apparatus Has Structural and Secretory Roles in Cyanobacterial Community Formation

Many roles of T4P in biofilm development in heterotrophic bacteria have been established, including patterning [93,94] and surface sensing [95,96]. In cyanobacteria, in contrast, research into T4P involvement in multicellular aggregate formation has been more limited. In *Synechocystis*, thick pili (via deletion of *pilA1*) and associated twitching motility were found to be non-essential for flocculation, whereas PilB1, the RNA chaperone Hfq, PilC, and the minor pilins of the *pilA9-slr2019* operon were crucial for flocculation and mutants in any of the respective genes lost flocculation entirely [71,73]. It has been suggested that some minor pilins may be incorporated into T4P as has been found in *P. aeruginosa* [97], with thin pili visible during electron microscopy potentially being composed of a different set of pilins. Neuhaus et al. recently found that *Thermus thermophilus* produces pili with different diameters that vary in pilin composition [18]. *Synechocystis* thin pili may thus likewise be (partially) composed of minor pilins and play an important role in flocculation.

T4P have been implicated in mechanosensing in several species of heterotrophic bacteria, sensing surface contact via downstream cAMP and c-di-GMP signalling and leading to enhanced surface colonisation [95,96,98]. There is currently no evidence that a similar process occurs in *Synechocystis* or other cyanobacteria, and neither PilA1 nor PilT1 being required for flocculation indicates that if mechanosensing via retractile T4P exists in *Synechocystis*, it is not essential for flocculation.

Biofilm formation in *S. elongatus* PCC 7942 suggests an additional, secretory role for the T4P apparatus [99]. Inactivation of the T4P apparatus leads to a biofilm phenotype in this cyanobacterium. Notably, wt conditioned medium was able to restore planktonic growth in these mutants, indicating that a small secreted factor inhibits biofilm formation in wt *S. elongatus* PCC 7942 [92]. In absence of this putative factor, short peptidase-processed proteins are secreted by *S. elongatus* PCC 7942 and support biofilm formation [100]. Although quorum sensing systems are widespread in proteobacteria and often essential for biofilm formation [101], limited examples of quorum sensing systems exist in cyanobacteria to date [102]. The suppression of *S. elongatus* PCC 7942 biofilm formation by secreted factors may represent a non-traditional quorum sensing system, suggesting quorum sensing may be more common in cyanobacteria than previously thought.

The type II secretion system (T2SS) is closely related to the T4P apparatus, with many T2SS protein components showing functions equivalent to their T4P counterpart. A schematic comparison between the architecture of T4P apparatus and the T2SS is shown in Figure 1. Although the *S. elongatus* PCC 7942 secretion ATPase deleted by Schatz et al. might be a homologue of either GspE or PilB [92], bioinformatic analyses have suggested that cyanobacteria do not contain a T2SS and the *S. elongatus* PCC 7942 system is rather a T4P system spread over multiple loci [103,104]. Denise et al. found that there are only two species (both *Gloeobacter*) containing generic type IV filament systems in the cyanobacterial clade and no dedicated T2SS at all, while type IVa pilus systems were found in 79 genomes, including various filamentous and unicellular cyanobacterial species [104]. Considering the ambiguity in assignment of the PilB-type ATPase responsible for the suppression of biofilm formation in *S. elongatus* PCC 7942, the impact of the *S. elongatus* PCC 7942 PilB homologue on biofilm formation and general protein secretion [99] suggests a secretory role of the T4P apparatus in biofilm formation in this cyanobacterium. Involvement of the T4P apparatus in secretion is not without precedent in cyanobacteria. Secretion of heterologous proteins in *Synechococcus elongatus* and other cyanobacteria has been suggested to proceed via the T4P apparatus by an unknown mechanism [105,106]. Furthermore, accumulation of extracellular PilA in *N. punctiforme* is connected with polysaccharide secretion via HmpF, PilB, and PilQ [6]. Cho et al. find that, particularly, loss of PilB and PilQ drastically reduces released polysaccharides and suggest polysaccharide export through the T4P machinery [6], though regulation via mechanosensing or similar T4P-mediated signalling mechanisms leading to downstream regulation of polysaccharide export by second messenger signalling could also explain the observations. Recently, several new genes have been identified in *N. punctiforme* encoding putative hormogonium polysaccharide (hps)-producing proteins, including homologues of Wza, Wzc, and Wzy [107]. Zuniga et al. also showed a connection between PilA secretion and the *hps* genes, with PilA secretion and motility depending strongly on several identified genes, although the mechanism of this interaction remains unclear [107]. In *Synechocystis*, polysaccharide export is substantially dependent on homologues of Wzm/Wzt [108] and Wza/Wzc [78], which have been found to be important in cell–cell and cell–surface adherence and cell buoyancy in *Synechocystis*, respectively. However, in contrast to *N. punctiforme*, no connection with the T4P machinery is known to date. This suggests that polysaccharide secretion may be dependent on T4P in some but not all species of cyanobacteria. Moreover, it has been shown that the homologue of the RNA chaperone Hfq binds to PilB1 in *Synechocystis* and that its correct localization at the pilus base is important for its function. Inactivation of *hfq* leads to non-motile cells and to changes in transcript accumulation, which are similar in *pilB1* and *pilC* mutant strains [109]. Hfq binds to a specific C-terminal domain of PilB1, which is found only in cyanobacterial assembly ATPases of T4P. Therefore, it is tempting to speculate that in general, in cyanobacterial mutants which do not form functional T4P, secondary effects occur, for example due to the incorrectly localised Hfq protein.

## 5. Regulation of the T4P Machinery

Although pilus motors can be directly regulated by c-di-GMP as found for the *Vibrio cholerae* Msh pilus [110], several other avenues of regulation exist, particularly at the transcriptional level. The second messenger cyclic 3’-5’-AMP (c-AMP) is induced by blue light in *Synechocystis* via the adenylate cyclase Cya1 [111], which is responsible for a large portion of intracellular cAMP production [112]. Which blue light-receptor is involved in activating Cya1 is currently unclear. Furthermore, *cya1* is downregulated by elevated bicarbonate levels [113], showing the diverse regulatory inputs in cAMP signalling. Cya1 is known to be crucial in *Synechocystis* phototaxis, with cells unable to form the characteristic finger-like projections in *cya1* deletion mutants [112,114]. Sycrp1 and Sycrp2 are the *Synechocystis* versions of the typical cAMP binding transcriptional regulator Crp known from many bacteria [115]. They propagate the Cya1 signal downstream, regulating among others the *pilA9-slr2019* operon and leading to a loss of cell surface piliation when deleted [116]. The downregulation of the *pilA9-slr2019* operon in Δ*sycrp2* mutants and the putative cooperation between Sycrp1 and Sycrp2 observed by Song et al. are sufficient to explain the impairment of motility in Δ*cya1*, Δ*sycrp1* [115], and Δ*sycrp2* [65,116] mutants. The abundance of *pilA11* mRNA and protein is further regulated by the PilR antisense RNA. PilR, however, does not regulate *pilA9*, *pilA10*, or *slr2018* expression, all part of the same operon as *pilA11*, showing that regulation of minor pilins can be highly specific [117].

Other species of cyanobacteria contain adenylate cyclases showing homology to *Synechocystis cya1* [112]. Though the connection to cell piliation in those bacteria is not as well established as in *Synechocystis*, cAMP signalling is involved in stress responses in other cyanobacteria such as desiccation tolerance in *Anabaena sp.* PCC 7120 [118], and mat formation in *Spirulina platensis*, both during extracellular cAMP addition [119] and when intracellular cAMP phosphodiesterases were inhibited [120]. Uptake of extracellular cAMP and compensation for low intracellular cAMP have also been observed in *Synechocystis* motility [114].

Several cyanobacteria have been confirmed to contain SigF, a stress response-related group 3 sigma factor, including many community-forming or filamentous species [121]. Although Imamura and Asayama found SigF in all cyanobacteria investigated, only *Synechocystis* and *N. punctiforme* SigF function has been characterised to date [121]. *Synechocystis* SigF is known to directly regulate the transcription of the *pilA1-pilA2* operon [122]. Deletion of *sigF* in *Synechocystis* correspondingly leads to a reduction in *pilA1-pilA2* mRNA and a loss of thick pili and phototactic motility [123,124]. Furthermore, Flores et al. found that *sigF* deletion significantly alters the exoproteome, with many proteins being present in reduced quantities or not at all [124]. Although this may indicate a connection between T4P action and protein secretion, as has been found in *S. elongatus* PCC 7942, Flores et al. also point out the large number of proteins of unknown function regulated by SigF, so definite conclusions on protein export will require further characterisation [124]. The *Synechocystis* Δ*sigF* mutant also shows an enhanced degree of cell sedimentation and flocculation [124], both in keeping with the drastic increase in exopolysaccharide production and change in monosaccharide composition. The increase in flocculation despite the downregulation of the *pilA1-pilA2* operon confirms that the major pilin PilA1 is not required in *Synechocystis* flocculation. Similarly, Miranda et al. have found *Synechocystis* to flocculate in batch cultures when the gene *slr1783*, coding for a monooxygenase, was deleted, leading to increased exopolysaccharide content [125]. The enhanced cellular aggregation in exopolysaccharide-overproducing mutants poses the question of the relative importance of T4P and exopolysaccharides in enabling *Synechocystis* flocculation depending on external conditions.

Similarly to *Synechocystis, N. punctiforme* SigF regulates *pilA*, which is very highly expressed in hormogonia, with deletion of *sigF* leading to almost total loss of *pilA* mRNA [126]. Intriguingly, all other T4P components investigated by Gonzalez et al. showed strong regulation by SigJ, rather than SigF [126]. The *Synechocystis sigF* deletion mutant showed no differential regulation of other components of the T4P [124], indicating that regulation of the major pilin specifically may be a common feature of cyanobacterial SigF.

## 6. Cyanobacterial Natural Competence Requires T4P

The uptake of exogenous DNA, known as natural competence, allows cells to adapt to environmental conditions through the exchange of situationally useful genes. Competence is known to depend on functional T4P in several bacteria [81,127], with DNA binding by T4P likely occurring at the pilus tip [128,129].

In *Synechocystis*, a close interplay between T4P and competence has also become evident. Yoshihara et al. have shown that many core components of the T4P apparatus are required for transformability in addition to their role in motility [9]. They also identified the *comA* gene, which, when deleted, causes a complete loss of transformability but only has a moderate impact on thick pili abundance and deletion mutants retain their motility [9]. Likewise, Nakasugi et al. identified a homologue of the *comF* gene in *Synechocystis*, which also leads to loss of competence but does not disrupt surface piliation when deleted [130]. *ComF* deletion, however, leads to a loss of motility [130]. This suggests a correlation between transformability and motility which was not apparent in the study by Yoshihara et al. on the effects of *comA* deletion. However, Yoshihara et al. assessed motility only from colony morphology, which may be less reliable than the colony motility assay method employed by Nakasugi et al, which is now more widely favoured. Nonetheless, these findings suggest that *Synechocystis* has DNA binding proteins which are required for natural transformability and which also strongly influence pilus function. Intriguingly, Nakasugi et al. also report the formation of filamentous aggregates in liquid cultures of *comF* deletion mutants matching the structures described in recent work on *Synechocystis* flocculation [71]. They suggest that the increased bundling of pili in the Δ*comF* mutant may be the cause of this flocculating phenotype, indicating that competence genes may be important factors in controlling *Synechocystis* T4P morphology and multicellularity [71].

Similar dependence of natural transformation on parts of the T4P apparatus has been observed in *S. elongatus* PCC 7942, including T4P motor proteins, *comEA*/*comEC*/*comF* homologues, and *sigF*, showing that many similar factors are involved as in *Synechocystis* [16]. Intriguingly, *S. elongatus* PCC 7942 is non-motile but naturally competent, asking questions about the cause of its deficiency in phototaxis, the mechanism of its competence, or both. Taton et al. also discovered a strong dependence of natural transformation on the circadian clock, likely via circadian control of the T4P machinery [16]. However, not all strains of *Synechococcus* are naturally competent, indicating some variability within the genus [131].

Although T4P in other naturally competent cyanobacteria have been suggested to be important in natural transformation, for example in *Microcystis aeruginosa* PCC 7806 [132], *Phormidium lacuna* [133], and *Thermosynechococcus elongatus* BP-1 [134], no direct link between T4P and competence in these organisms has been made to date. Recent in silico approaches, however, have shown that many cyanobacteria share high sequence similarity in their T4P genes with naturally competent cyanobacteria [16,133], suggesting that T4P may enable natural transformation in many more cyanobacterial genera than have been confirmed experimentally.

## 7. Concluding Remarks

Several unanswered questions on the involvement of T4P in cyanobacterial motility and multicellularity remain. Particularly mechanosensing, emerging as a frequent feature among heterotrophic bacteria, has not yet been demonstrated in cyanobacteria. Aggregation into biofilms, flocs, and microbial mats is a more common strategy in cyanobacteria than previously thought, particularly among laboratory strains such as *Synechocystis* or *S. elongatus* PCC 7942. The latter strain provides an example of the dangers of relying on strains that exhibit potentially non-representative phenotypes with the isolation of *S. elongatus* UTEX 3055, a very close relative of *S. elongatus* PCC 7942, indicating that freshwater *Synechococcus* may be motile and natively community-forming. *Thermosynechococcus vulcanus* NIES-2134 (RKN) and *Thermosynechococcus elongatus* BP-1 similarly are very closely related [135] and yet show divergent aggregation phenotype, with *T. elongatus* BP-1 being deficient in aggregation [136], illustrating the importance of selecting appropriate strains for research. The examples of *S. elongatus* PCC 7942 and *N. punctiforme* have demonstrated that much is left to be understood about the secretory roles of the T4P apparatus in cyanobacteria, providing potential avenues of cell–cell communication which have so far been largely missing in the phylum.

The loss of motility in laboratory strains as a result of prolonged cultivation in particular is further exemplified by the microevolution of *Synechocystis* laboratory strains [137]. We have likewise observed in the past that at least some strains of the non-motile Kazusa branch of *Synechocystis* (ATC27184) do not flocculate. It therefore seems prudent to be mindful of such pitfalls given the large areas of cyanobacterial physiology that are yet to be thoroughly explored.

Research in the last few years has shown that cyanobacteria are capable of complex and co-operative behaviour. Much remains to be learned about these behaviours and the survival advantages that they may confer in the natural environment.

## Figures and Tables

**Figure 1 life-10-00252-f001:**
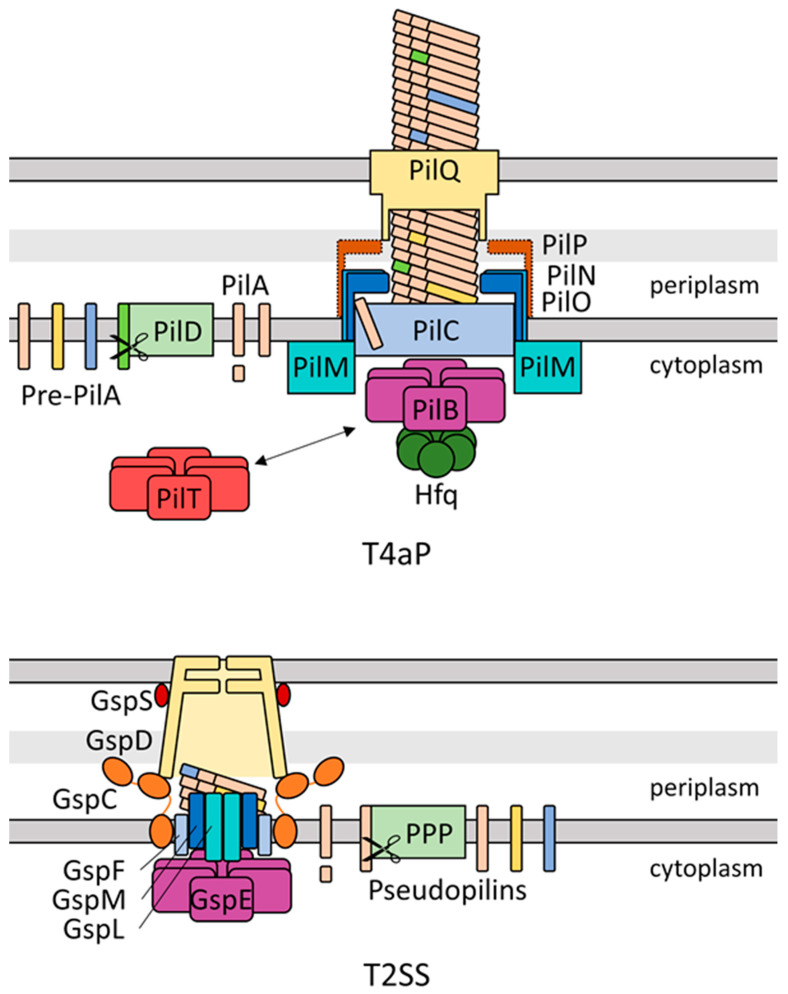
Schematic representation of the *Synechocystis* Type IVa pilus (T4aP) apparatus adapted from *Myxococcus xanthus* structure and nomenclature [12] and of the type II secretion system (T2SS) based on structural data from [13,14,15] using Escherichia component nomenclature. Colours denote proteins fulfilling homologous functions between the two systems. Dotted outline of PilP denotes a lack of experimental data confirming the in silico identification by Taton et al. [16].

**Figure 2 life-10-00252-f002:**
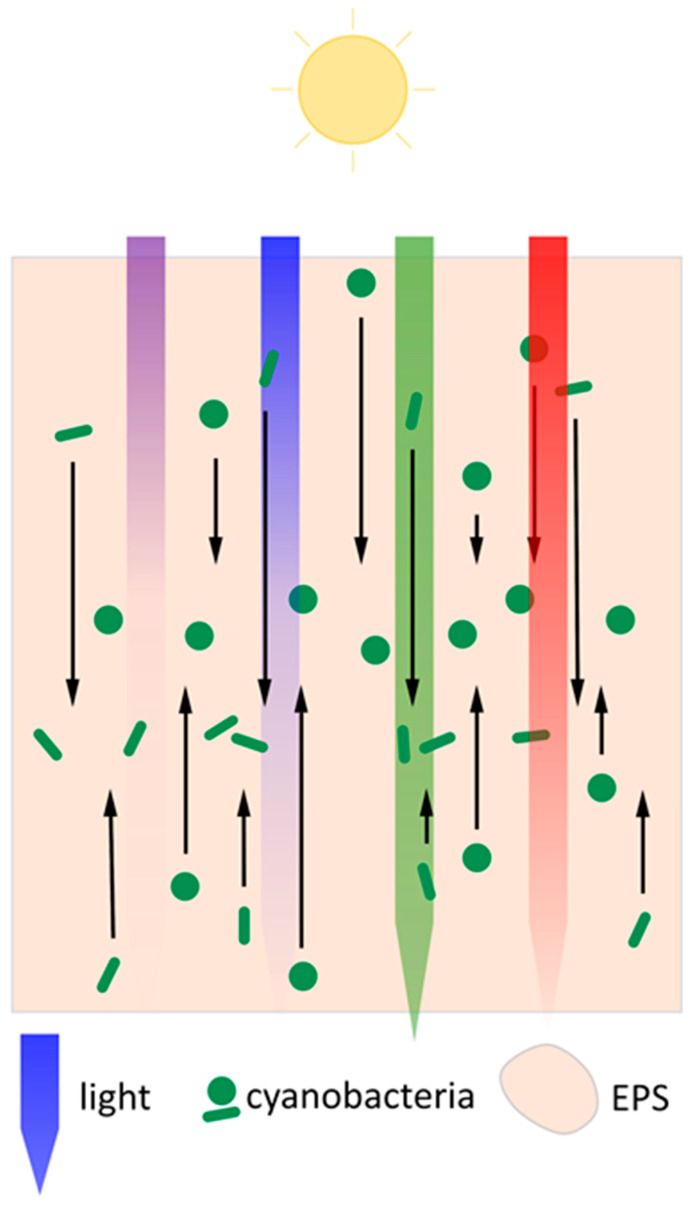
Model of movement of single-celled cyanobacteria in a multi-species phototrophic microbial mat. Black arrows indicate twitching motility in the direction of the arrow. The relative attenuation of different wavelengths of light in phototrophic communities (adapted from [67]) is represented by the colour gradients of the respective downward arrows, with purple representing UV-A radiation. The distinct light environments at different depths in the mat can activate or inactivate motility and trigger switches from positive to negative phototaxis and vice versa. This leads to an accumulation of the cyanobacteria at a depth where there is a favourable light environment.

**Figure 3 life-10-00252-f003:**
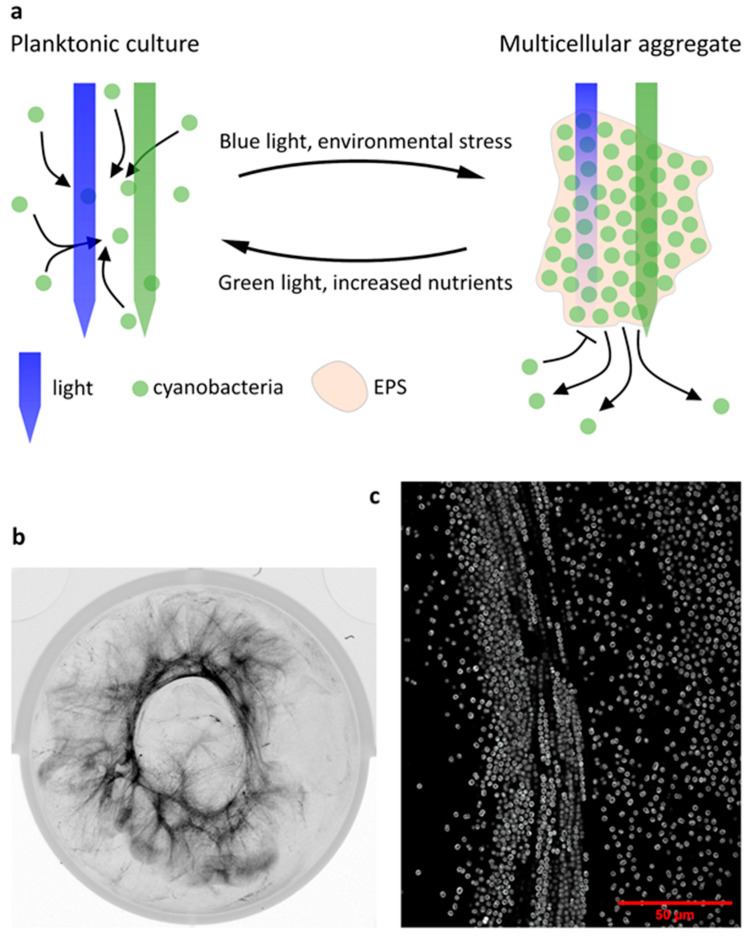
(**a**) Model of dynamic aggregation in *Synechocystis* and the *Thermosynechococcus* genus depending on external factors such as light penetration. Small arrows denote aggregation or the dissolution of aggregates. (**b**) Example of *Synechocystis* wt flocculation of a liquid culture in a 6-well plate (imaged vertically) with autofluorescence displayed in inverted greyscale. (**c**) Confocal micrograph of a *Synechocystis* floc, displaying autofluorescence in greyscale.

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
