# Peer review of "The Role of the Cyanobacterial Type IV Pilus Machinery in Finding and Maintaining a Favourable Environment"

_life, 2020, doi:10.3390/life10110252_

Round 1
Reviewer 1 Report
The review by Conradi et al. is an interesting and timely document summarizing current knowledge of cyanobacterial type 4 pili (T4P) assembly and functions. This review gathers information, from earlier studies through up-to-date publications, and comprehensively covers several subjects of interest. A major focus is given to phototaxis, the movement towards or away from light. The authors thoroughly describe the mechanism and its regulation and provide summarizing models describing the benefits of this phenomenon, e.g. optimal cell positioning in a water column or within a community such as a microbial mat. Additionally, role of motility in formation of multicellular assemblages is described, as well as roles of the T4P apparatus in secretion and DNA competence. The review is tightly and clearly written and it certainly merits publication. Below are a few minor comments for the authors' consideration.
- Lines 105-107: "Light perception in Phormidium uncinatum, …. depends on decreases of light at the leading end of the filament and increases in light at the lagging end, though the mechanism for this spatial comparison is unclear." The description decreases of light and increases of light in this context is very unclear. Please revise to clarify.
- Reference to the study indicating involvement of a PilB homologue in protein secretion (currently line 259) better fits the section entitled “The T4P apparatus has structural and secretory roles in cyanobacterial community formation”.
- The authors may consider adding a section entitled "Concluding remarks/Future directions.
Author Response
First, let us thank the reviewer for the very valuable comments. We have comprehensively revised the manuscript in response to the comments of all three reviewers. You will find the comments of reviewer 1 below, followed by our answers in bold.
Reviewer 1:
- Lines 105-107: "Light perception in Phormidium uncinatum, …. depends on decreases of light at the leading end of the filament and increases in light at the lagging end, though the mechanism for this spatial comparison is unclear." The description decreases of light and increases of light in this context is very unclear. Please revise to clarify.
Thank you for pointing this out. We have revised our statement (see line 107-109).
- Reference to the study indicating involvement of a PilB homologue in protein secretion (currently line 259) better fits the section entitled “The T4P apparatus has structural and secretory roles in cyanobacterial community formation”.
We have now shifted this as suggested to section 4 (see lines 320-323)
- The authors may consider adding a section entitled "Concluding remarks/Future directions.
Thank you for this valuable comment. We followed also the suggestion of reviewer 3 to include parts of our last paragraphs into this section.
Reviewer 2 Report
This review manuscript is dealing with type IV pilus machinery in cyanobacteria, including various aspects such as physiology and regulation. This manuscript is well organized and the English are very nice. Thus, the review manuscript should largely contribute to the cyanobacterial scientific community, and so be suitable for publication after several concerns have been solved.
Major Comments
1. In section 2, the authors summarized phototactic regulation mainly based on the studies on S. 6803. The authors described UirS, PixJ1, and Cph2, but not PixD, the BLUF-based blue light sensor in cooperation with PixE. I recommend incorporating PixD-PixE stories. Furthermore, not only PixG but also LsiR and PixE are response regulators containing PATAN domain at the N-terminus, which should be described in the context of possible common output to the T4P apparatus. While PixJ and its gene cluster are highly conserved among various cyanobacteria, UirS, Cph2, and PixD are not so conserved, indicating that the PixJ signaling may be the main regulatory path for the phototactic regulation in cyanobacteria. Further, sensing light qualities of the GAF domains of the PixJ homologs are highly diversified: not only blue/green photocycle but also red/green and many others. I recommend to briefly describe these things in this section to cite several references especially the Nostoc PtxD (NpF2164) Ref (Campbell et al. J. Bacterial. 2015 197 (4): 782-791 & Rockwell et al. 2012 Biochemistry 51 (48): 9667-9677).
2. L231-L232: Enomoto's works, Ref. 75 & 76, both deal with T. vulcanus aggregation. T. elongatus did NOT show any aggregation under low-temperature light conditions, whereas T. vulcanus well shows the cell aggregation. So, Enomoto and co-authors focused on T. Vulcanus physiology, although their initial in vitro biochemical works on SesA has been focused on T. elongatus. Anyway, the authors should carefully read these references and describe them correctly. This physiological difference between T. elongatus and T. vulcanus is analogous to the issues of Synechococcus and Synechocystis microevolution described at L412-427. Thermosynechococcus issues may be also incorporated in this section.
Minor Comments
3. L65 There are many PilA homologs in S. 6803 and N. punctiforme. It is preferable to describe whether the PilA homologs of the other cyanobacteria and bacteria are also redundant or not.
4. L143: flavin -> flavin mononucleotide
Author Response
First, let us thank the reviewer for the very valuable comments. We have comprehensively revised the manuscript in response to the comments of all three reviewers. You will find the comments of reviewer 2 followed by our answers in bold
- In section 2, the authors summarized phototactic regulation mainly based on the studies on S. 6803. The authors described UirS, PixJ1, and Cph2, but not PixD, the BLUF-based blue light sensor in cooperation with PixE. I recommend incorporating PixD-PixE stories.
We now include a paragraph (lines 162-166) on the involvement of the PixD-PixE system in phototaxis.
Furthermore, not only PixG but also LsiR and PixE are response regulators containing PATAN domain at the N-terminus, which should be described in the context of possible common output to the T4P apparatus.
We have incorporated the reviewer’s suggestion into lines 167-169.
While PixJ and its gene cluster are highly conserved among various cyanobacteria, UirS, Cph2, and PixD are not so conserved, indicating that the PixJ signaling may be the main regulatory path for the phototactic regulation in cyanobacteria. Further, sensing light qualities of the GAF domains of the PixJ homologs are highly diversified: not only blue/green photocycle but also red/green and many others. I recommend to briefly describe these things in this section to cite several references especially the Nostoc PtxD (NpF2164) Ref (Campbell et al. J. Bacterial. 2015 197 (4): 782-791 & Rockwell et al. 2012 Biochemistry 51 (48): 9667-9677).
We thank the reviewer for this suggestion. We have added information on some cyanobacterial PixJ homologues (lines 151-157) but feel that more detailed exploration of PixJ photocycles would be beyond the scope of this review which mainly focusses on the involvement of type IV pili in cellular behaviour.
- L231-L232: Enomoto's works, Ref. 75 & 76, both deal with T. vulcanus aggregation. T. elongatus did NOT show any aggregation under low-temperature light conditions, whereas T. vulcanus well shows the cell aggregation. So, Enomoto and co-authors focused on T. Vulcanus physiology, although their initial in vitro biochemical works on SesA has been focused on T. elongatus. Anyway, the authors should carefully read these references and describe them correctly. This physiological difference between T. elongatus and T. vulcanus is analogous to the issues of Synechococcus and Synechocystis microevolution described at L412-427. Thermosynechococcus issues may be also incorporated in this section.
Thank you for this very important comment and we apologise for the incorrect citation of Enamoto’s work in our initial submission. We have modified the section accordingly. In addition, we have discussed the physiological differences between both Thermosynechococccus strains and microevolution with Gen Enomoto personally. His suggestion was to omit this from the discussion because currently there is no clear explanation for the different aggregation phenotypes. However, we included a short statement on this in the concluding remarks (line 445-447).
Minor Comments
- L65 There are many PilA homologs in S. 6803 and N. punctiforme. It is preferable to describe whether the PilA homologs of the other cyanobacteria and bacteria are also redundant or not.
We are not sure what the reviewer requires here. We made no comment at this point about the redundancy or otherwise of the multiple PilA homologs. Recent research suggests specialised functions for at least some of the pilin homologs (which are usually called minor pilins because of their lower abundance), as we discuss later in the context of flocculation, for example. Therefore, it is not possible to conclude about a redundancy independent of specific functions of the minor pilins.
- L143: flavin -> flavin mononucleotide
Has been changed.
Reviewer 3 Report
The review entitled “The role of the cyanobacterial type IV pilus machinery in finding and maintaining a favourable environment” by Conradi, Mullineaux and Wilde describes several established aspects of cyanobacterial motility using type IV pilus assemblies and expands previous research covering cyanobacterial motility with the newest experimental observations, adding information on so far only sparsely described cyanobacterial cell-cell aggregation mechanisms. This well written and comprehensive review gives rise to only few comments from my side; while in most instances phenomena are described in detail, in some places, a bit more in depth background information would be desirable for readers not entirely familiar with the topic of cyanobacterial motility. In recommend this review for publication as it provides a valuable summary for cyanobacterial researchers and adds intriguing future research perspectives. I would, however, ask for a few minor points to be addressed before publication that are listed below:
Line 43: I would suggest writing “pilin proteins” instead of pilins.
Line 48: In Figure 1, the first figure is described as T4aP, shouldn’t it be T4P here? If not, what does the “a” stand for? In this context, in the first mentioning of a figure in the text, the authors decide to use a capital “F”, while in the subsequent mentioning of figures, figure is always written with a lower-case “f”. Please unify.
Line 60-62: One reference at the end of the sentence would be enough.
Line 63: Please spell in full the species name for T. thermophilus.
Line 95: Please provide a reference.
Line: 108: I assume the authors mean Anabaena variabilis?
Line 118: Given its likely non-chromosome maintenance function, the protein is likely better characterized as a coiled-coil-rich protein instead of an SMC protein.
Line 141: Please write out in full “LOV” and “BLUF”.
Line 170: From this paragraph the connection of photoreceptors and multicellularity is not clear. Please either delete multicellularity here or give an example of such dependency.
Line 175: Biofilms were not mentioned before, why do the authors decide to give a model in this context here? Maybe the reference to Figure 2 works better in the next paragraph where biofilms are discussed or briefly mention why a biofilm is a particular good example (which it is) for phototactic behavior.
Line 189: I would argue that cell-cell assemblies are no true displays of bacterial multicellularity. This also applies to some other lines further below in the manuscript. Given that some previous literature agrees with the authors description of multicellularity (e.g. Claessen et al., 2014), I could understand if the authors prefer to leave it as it is. I would, however, rather describe those assemblies as transient multicellular. They lack stable cell-cell contacts and direct cell-cell communication via pores/channels and they do not differentiate specialized cells.
Line 205-208: Is this hypothesis from the current manuscript or from reference 6? I cannot find any mentioning of surface-area-to-volume ratios or related aspects in reference 6.
Line 223: Please provide a brief description of what Fig. 3b shows. Are we looking at a culture in a well or on a growth plate or something entirely different?
Line 228: I think it would be helpful to provide a brief description/definition of biofilms vs. flocs.
Line 246: A comma after knowledge.
Line 250-252: Please provide a reference.
Line 295-298: What is the difference between type IV and type Iva pili? Please briefly describe.
Line 300: “implies” is a strong word here, maybe “suggests” or “indicates” would fit better, given that the authors indicate a few lines below that other explanations also exist.
Line 385-388: Why is it surprising that Yoshihara’s findings on comA are different than the findings from Nakasugi on comF? Those are two different genes that can have different phenotypes upon deletion.
Line 390-391: It would be helpful to indicate those proteins in Fig. 1 and refer to the figure here or otherwise describe how they are pilus-based (i.e., are they directly interacting with some of the mentioned pilus proteins or such?).
Line 412-427: I think those two paragraphs would make up a nice independent caption (i.e. 7. Outlook).
Line 424: Remove the bracket after the reference.
Author Response
Thank you for the very valuable comments. We have comprehensively revised the manuscript in response to the comments of all three reviewers. You will find the comments of reviewer 3 followed by our answers in bold.
Line 43: I would suggest writing “pilin proteins” instead of pilins.
Agreed, we have changed it here to “pilin proteins”. However, we still have used the term “pilins” in the manuscript because this term is used very frequently in the field.
Line 48: In Figure 1, the first figure is described as T4aP, shouldn’t it be T4P here? If not, what does the “a” stand for? In this context, in the first mentioning of a figure in the text, the authors decide to use a capital “F”, while in the subsequent mentioning of figures, figure is always written with a lower-case “f”. Please unify.
Thank you for this important comment. We have changed this part of the manuscript accordingly (lines 40-44) and introduce now the type Iva pilus system.
Line 60-62: One reference at the end of the sentence would be enough.
We changed it.
Line 63: Please spell in full the species name for T. thermophilus.
We changed it.
Line 95: Please provide a reference.
We have introduced reference 24.
Line: 108: I assume the authors mean Anabaena variabilis?
Yes, that’s correct, we have changed it.
Line 118: Given its likely non-chromosome maintenance function, the protein is likely better characterized as a coiled-coil-rich protein instead of an SMC protein.
Thank you for this comment, we have changed it accordingly (line 120).
Line 141: Please write out in full “LOV” and “BLUF”.
Done.
Line 170: From this paragraph the connection of photoreceptors and multicellularity is not clear. Please either delete multicellularity here or give an example of such dependency.
We have clarified this point by rephrasing to ‘biofilm formation’ (line 185). The intent of this statement was not to claim direct dependency but rather suggest the possibility of such as might be expected based on the phosphodiesterase activity of SL2, and that such studies would benefit from the use of the motile and natively biofilm-forming S. elongatus UTEX 3055 strain.
Line 175: Biofilms were not mentioned before, why do the authors decide to give a model in this context here? Maybe the reference to Figure 2 works better in the next paragraph where biofilms are discussed or briefly mention why a biofilm is a particular good example (which it is) for phototactic behavior.
Agreed, we have put a reference to figure 2 now in the next paragraph (line 190) and added a sentence (line190-192) on biofilm formation and its possible connection to phototaxis response.
Line 189: I would argue that cell-cell assemblies are no true displays of bacterial multicellularity. This also applies to some other lines further below in the manuscript. Given that some previous literature agrees with the authors description of multicellularity (e.g. Claessen et al., 2014), I could understand if the authors prefer to leave it as it is. I would, however, rather describe those assemblies as transient multicellular. They lack stable cell-cell contacts and direct cell-cell communication via pores/channels and they do not differentiate specialized cells.
According to your suggestion, we have removed “multicellular“ from the heading to avoid any confusion (line 204). The subsequent text explains what we mean here by “multicellular“, which is not intended to necessarily imply stable cell-cell contacts or direct cell-cell communication.
Line 205-208: Is this hypothesis from the current manuscript or from reference 6? I cannot find any mentioning of surface-area-to-volume ratios or related aspects in reference 6.
The hypothesis that filamentous floc structure enables greater nutrient penetration into flocs is briefly discussed in Conradi et al., 2019 (originally reference 61; now reference 71).
Line 223: Please provide a brief description of what Fig. 3b shows. Are we looking at a culture in a well or on a growth plate or something entirely different?
We have added now a more detailed description (lines 241-242).
Line 228: I think it would be helpful to provide a brief description/definition of biofilms vs. flocs.
We have added a sentence to better define flocs (lines 222-223).
Line 246: A comma after knowledge.
Done.
Line 250-252: Please provide a reference.
We have inserted reference 89 and 37 for the Thermosynechococcus and Synechocystis work, respectively (line 262-263).
Line 295-298: What is the difference between type IV and type Iva pili? Please briefly describe.
We describe this now in lines 40-44.
Line 300: “implies” is a strong word here, maybe “suggests” or “indicates” would fit better, given that the authors indicate a few lines below that other explanations also exist.
Changed (line 299).
Line 385-388: Why is it surprising that Yoshihara’s findings on comA are different than the findings from Nakasugi on comF? Those are two different genes that can have different phenotypes upon deletion.
Yes, we agree, we have rephrased the section in question to clarify (lines 405-411).
Line 390-391: It would be helpful to indicate those proteins in Fig. 1 and refer to the figure here or otherwise describe how they are pilus-based (i.e., are they directly interacting with some of the mentioned pilus proteins or such?).
Though these proteins have been indicated in come illustrations, we think that the mechanism of action and potential interaction partners of ComA and ComF in cyanobacteria is very unclear and thus we suggest not to include them in this review in figure 1. Instead, we have modified lines 412-413 to better reflect the intended meaning.
Line 412-427: I think those two paragraphs would make up a nice independent caption (i.e. 7. Outlook).
Yes, we have changed this now. See new Paragraph 7. Concluding remarks.
Line 424: Remove the bracket after the reference.
Done.